# How do authors' perceptions of their papers compare with co-authors' perceptions and peer-review decisions?

Charvi Rastogi[1], Ivan Stelmakh[2], Alina Beygelzimer[3], Yann N. Dauphin[4], Percy Liang[5], Jennifer Wortman Vaughan[6], Zhenyu Xue[7], Hal Daumé III[8], Emma Pierson[9], Nihar B. Shah[1]*

1 Machine Learning Department, Carnegie Mellon University, Pittsburgh, Pennsylvania, United States of America, 2 New Economic School, Moscow, Russia, 3 Yahoo! Research, New York, New York, United States of America, 4 Google Deepmind, Montreal, Canada, 5 Department of Computer Science, Stanford University, Stanford, California, United States of America, 6 Microsoft Research, New York, New York, United States of America, 7 Independent Researcher, Shanghai, China, 8 Department of Computer Science, University of Maryland, College Park, Maryland, United States of America, 9 Jacobs Technion-Cornell Institute, Cornell Tech, New York, New York, United States of America

* nihars@cs.cmu.edu

**Data Availability Statement:** In this work, we analyze data obtained from the scientific conference NeurIPS (Conference on Neural Information Processing Systems) 2021 for

## Abstract

How do author perceptions match up to the outcomes of the peer-review process and perceptions of others? In a top-tier computer science conference (NeurIPS 2021) with more than 23,000 submitting authors and 9,000 submitted papers, we surveyed the authors on three questions: (i) their predicted probability of acceptance for each of their papers, (ii) their perceived ranking of their own papers based on scientific contribution, and (iii) the change in their perception about their own papers after seeing the reviews. The salient results are: (1) Authors had roughly a three-fold overestimate of the acceptance probability of their papers: The median prediction was 70% for an approximately 25% acceptance rate. (2) Female authors exhibited a marginally higher (statistically significant) miscalibration than male authors; predictions of authors invited to serve as meta-reviewers or reviewers were similarly calibrated, but better than authors who were not invited to review. (3) Authors' relative ranking of scientific contribution of two submissions they made generally agreed with their predicted acceptance probabilities (93% agreement), but there was a notable 7% responses where authors predicted a worse outcome for their better paper. (4) The author-provided rankings disagreed with the peer-review decisions about a third of the time; when co-authors ranked their jointly authored papers, co-authors disagreed at a similar rate—about a third of the time. (5) At least 30% of respondents of both accepted and rejected papers said that their perception of their own paper improved after the review process. The stakeholders in peer review should take these findings into account in setting their expectations from peer review.

publication of research in AI. For this, we worked in collaboration with the conference organizers of Neurips 2021 (Alina Beygelzimer, Yann N. Dauphin, Percy Liang, Jennifer Wortman Vaughan). We are not able to share this data in any form, because of the following reasons: 1. This is highly sensitive data concerning authors and reviewers both, where authors and reviewers provide their honest opinion with the belief that this would not affect them personally. 2. NeurIPS 2021 is a double-blind conference and maintaining the double-blindness is an important aspect of the reviewing process. Releasing de-identified data would dilute the double-blindness. 3. It would violate the confidentiality agreement between the researchers who participated in the conference and the conference organizers. We are only able to share the aggregated statistics, which are provided in the manuscript. Further, any data requests from researchers who meet the criteria for access to confidential data may be sent to the Neural Information Processing Systems foundation that is the institutional body responsible for running the conference NeurIPS 2021, or to the Institutional Review Board of Microsoft Research (contact via MSRethics@microsoft.com).

**Funding:** The author(s) received no specific funding for this work.

**Competing interests:** The authors have declared that no competing interests exist.

# 1 Introduction

Peer review is used widely in scientific research for quality control as well as selecting 'interesting' research. However, a number of studies have documented low agreement among reviewers [1–7], and researchers often lament various problems with peer review [8–10]. On the other hand, surveys of researchers about their general perception of peer review reveal that researchers across various scientific disciplines consider peer review to be important, yet in need of improvements [11–15]. But how do author perceptions on their submitted papers match up with outcomes of the peer-review process? We investigate this question in this work.

We conducted a survey-based experiment in the Neural Information Processing Systems (NeurIPS) 2021 conference, which is a top-tier conference in the field of machine learning. (Readers outside computer science unfamiliar with its publishing culture may note that in computer science, conferences review full papers and are commonly the terminal venue of publication of papers). The conference had over 9,000 papers submitted by over 23,000 authors. The conference traditionally has accepted 20–25% of the submitted papers, and in 2021 the acceptance rate was 25.8%.

We designed and executed a survey to understand authors' perceptions about their submitted papers as well as their perceptions of the peer-review process in relation to their papers. In particular, we asked three questions:

- It is well known that the peer-review process (at the NeurIPS conference) has a low acceptance rate and a high amount of disagreement between reviewers [3, 5, 16]. Do authors take this into account when setting their expectations from the peer review process? Specifically, we aim to understand the calibration of authors with respect to the review process, by asking them to predict the probability of acceptance of their submitted paper(s).

- Motivated by authors often lamenting that their paper that they thought was best was rejected and the one they thought had lower scientific merit was accepted, we aim to quantify the discrepancy between the author's and the reviewers' relative perceptions of papers by asking authors to rank their papers in terms of their perceived scientific contribution and comparing this against acceptance decisions.

- Finally, while the two questions above measured the perception before the review process, we also measured the perception after they see the reviews, by asking authors whether the review process changed their perception of their own paper.

We then analyze how author perceptions align with the outcomes of the peer-review process and the perceptions of co-authors. The results of this work are useful to set expectations from the peer-review process, identify its fundamental limitations, and help guide the policies that the community implements as well as future research on improving peer review.

The rest of the paper is organized as follows. Section 2 discusses related work. In Section 3, we present details of the questions asked to the participants (authors of submitted papers). We provide basic statistics of the responses in Section 4 and our main analysis of the responses in Section 5. We conclude with a discussion in Section 6.

# 2 Related work

There are a number of papers in the literature that conduct surveys of authors. [17] survey authors of *accepted* papers from 56 computer science conferences. The survey was conducted after these papers were published. Questions pertained to the paper's history (amount of time needed to write it; resubmission history) and their opinions about the conference's rebuttal process. The respondents were also asked whether they found the reviews helpful in improving

their paper. A total of 34.1% of the respondents said they were 'very helpful,' 52.7% said they were 'somewhat helpful,' and 13.2% said they were 'not at all' helpful. Similar surveys asking authors whether peer review helped improve their paper are also conducted in other fields [11, 18, 19]. It is important to note that this question is different from our third question which asks whether their own perception of the quality of their own paper changed after the review process. Our question pertains to the same (version of the) paper but perception before and after the reviews; on the other hand, their question pertains to two different versions of the paper (initial submission and after reviewers' suggestions) and whether there was an improvement across the versions. They also find that for these questions, responses from different authors to the same paper were usually very similar.

[20] surveys authors of research proposals on their perception of random allocation of grant funding. They do find support for such randomized decisions, which have now also been implemented [21]. [22, 23] conduct or analyze surveys of authors for breach of ethics. While computer science was not their focus, within computer science as well, there have been discoveries of breach of ethics in the peer-review process [24–28].

Several other surveys [11–15] find a strong support for peer review among researchers. They also find that researchers see a need to improve peer review.

The work of [29] is closest to ours. They conduct a survey in the Australasian Association for Engineering Education (AAEE) annual conference 2010 and 2011, comprising a total of 70 papers and 140 reviews. The survey asked authors to rate their own papers and also to rate reviews. Their survey received responses from 23 authors in 2010 and from 37 authors in the 2011 edition. They found that overall 75% of authors rated their paper higher than the average of the reviewers' ratings for their paper. Furthermore, their survey found that the academic rank of the respondent was not correlated with the accuracy of the respondent's prediction of the reviews.

[30] offers a somewhat tongue in cheek commentary pertaining to authors' perceptions: *"if authors systematically overestimate the quality of their own work, then any paper rejected near the threshold is likely to appear (to the author) to be better than a large percentage of the actual conference program, implying (to the author) that the program committee was incompetent or venal. When a program committee member's paper is rejected, the dynamic becomes self-sustaining: the accept threshold must be higher than the (self-perceived) merit of their own paper, encouraging them to advocate rejecting even more papers."*

Within the machine learning community, [31] survey reviewers about visibility of papers submitted to a conference that anonymizes authors, and intentionally searching online for assigned papers. Or current work contributes to a tradition in machine learning venues of experimentation aimed at understanding and improving the peer-review process [3, 5, 16, 32–38].

See [39] for a more extensive discussion about research on the peer-review process and associated references.

## 3 Methods

Our experiment was conducted in two phases. Phase 1 was conducted shortly after the paper submission deadline, and Phase 2 was conducted after the authors were shown their initial reviews. (During the NeurIPS 2021 review process, initial reviews were released to authors, who had the chance to respond to the reviews and engage in subsequent discussion with the reviewers. Reviews were then updated before final acceptance decisions were released). We asked two questions during Phase 1 and one question during Phase 2, as described below. All of the questions were optional. Authors were told that their responses will not be seen by anyone during the review process and will not affect the decisions on

their papers. The study protocol was approved by an independent institutional review board (IRB). A more detailed description of privacy and confidentiality of responses can be found in Appendix A.

**Phase 1**: The first phase was conducted four days after the deadline for submission of papers, and was open for ten days. All authors of submitted papers were asked the following question:

- **Acceptance probability**. What is your best estimate of the probability (as a percentage) that this submission will be accepted? Please use a scale of 0 to 100, where 0 = "no chance of acceptance" and 100 = "certain to be accepted." Your estimate should reflect only how likely you believe it is that the paper will be accepted at NeurIPS, which may or may not reflect your perception of the actual quality of the submission. For context, over the past four years, about 21% of NeurIPS submissions were accepted.

Every author who had authored more than one submitted paper was also asked the following second question:

- **Paper quality ranking**. Rank your submissions in terms of your own perception of their scientific contributions to the NeurIPS community, if published in their current form. Rank 1 indicates the submission with the greatest scientific contribution; ties are allowed, but please use them sparingly.

Notice that the two questions differ in two ways. The acceptance probability question asks for a value (chance of acceptance), and this value represents the authors' perception of the outcomes of the peer-review process for their paper. On the other hand, the paper quality ranking question asks for a ranking, and furthermore, pertains to the author's own perception of the scientific contribution made by their paper.

**Phase 2**: The second phase was conducted after the authors could see the (initial) reviews. This phase comprised a single question, and the participants were told they could answer this question irrespective of whether they participated in Phase 1 or not.

- **Change of perception**. After you read the reviews of this paper, how did your perception of the value of its scientific contribution to the NeurIPS community change (assuming it was published in its initially submitted form)? [Select any one of the following options.]

  - My perception became much more positive

  - My perception became slightly more positive

  - My perception did not change

  - My perception became slightly less positive

  - My perception became much less positive

More details about the timeline and instructions are provided in Appendix A. The instructions were designed to give participants complete information about how their provided data would be used. The participant emailing and data collection for the experiment started on June 1, 2021 and ended on September 28, 2021.

### 3.1 Ethics statement

The experiment was reviewed and approved as exempt research by the Microsoft Research IRB, and the data collected were analysed anonymously.

## 4 Basic statistics

In this section, we provide some basic statistics pertaining to the experiment.

### 4.1 NeurIPS 2021 conference

- Total number of papers submitted to the conference: 9,034.

- Total number of unique authors who submitted papers to the conference: 23,882.

- Total number of author-paper pairs: 37,100. (Only authors with a profile on the conference management platform (OpenReview.net) could participate in the experiment, yielding 34,713 eligible author-paper pairs).

- Percentage of submitted papers that were eventually accepted to the conference: 25.8%.

We now move on to discuss the responses to the three questions.

### 4.2 "Acceptance probability" question

- Number of responses: 9,907 (26.7% of author-paper pairs).

- Number of papers with at least one response: 6,278 (69.5%).

### 4.3 "Paper quality ranking" question

- Number of authors with more than one submission: 6,237.

- Total number of author-paper pairs for these authors: 19,455.

- Number of "rank" responses received (out of 19,455): 6,908 (35.5% response rate).

### 4.4 "Change of perception" question

- Number of papers remaining after reviews were released (as some were rejected/withdrawn): 8,765

- Number of author-paper pairs remaining: 36,103.

- Number of responses: 4,435 (12.3% response rate).

**4.4.1 Response rates and breakdown.** The overall response rates in our experiment are broadly in the ballpark of the response rates of other surveys in computer science. The survey by [40] in the CHI 2011 conference had a response rate of 16%. [31] conduct multiple surveys: an anonymous survey in the ICML 2021 and EC 2021 conferences had response rates of 16% and 51% respectively; a second, non-anonymous opt-in survey in EC 2021 had a response rate of 55.78%. [17] survey authors of accepted papers in 56 computer systems conferences, with response rates ranging from 0% to 59% across these conferences. The survey by [29] was opt-in in 2011 and their response rate was 28%.

We used gender self-reported in OpenReview profiles. The conference had 23,581 author-paper pairs with a self-reported gender "male" of the author, 3,328 author-paper pairs with a self-reported gender "female" of the author. We omit other gender-based subgroups in our analysis, following concerns about privacy and noise due to the small sample size of responses from these groups. Further, 7,432 author-paper pairs did not have a self-reported gender of the

author. In phase 1, the response rate among author-paper pairs with self-reported gender as "male" was 30.9%, that among self-reported gender as "female" was 24.7%, and among the rest was 22%.

In terms of seniority, while we do not have a perfect measure of seniority, we used the role within the NeurIPS 2021 reviewing process as a proxy. We consider three levels of seniority. Ordered by decreasing seniority, these levels comprise of: (1) authors who were invited to serve as area chairs or senior area chairs at NeurIPS 2021, whom we refer to as "meta-reviewers"; (2) authors who were invited to serve as reviewers; and (3) authors who were in neither of the two aforementioned groups. The conference saw 3,834 author-paper pairs for authors invited as meta-reviewers, 10,938 pairs for authors invited as reviewers, and 19,941 for those who were in neither list. The response rate (for acceptance probability) was 21% among authors invited as meta-reviewers, 28.9% among authors invited to review, and 29.8% for those in neither list.

In terms of paper outcomes, out of all the responses to Phase 1 (acceptance probability) of the experiment, 27% of the responses pertained to papers that were eventually accepted. Thus, in Phase 1 we did not see any large non-response bias with respect to the papers that were eventually accepted or rejected.

The "change of perception" question (Phase 2) was asked after the authors saw the reviews. Some authors with unfavorable reviews withdrew their papers before this phase. Some other papers were rejected before this phase for reasons such as formatting violations. As a result, out of all the responses to Phase 2 of the experiment, there was a significantly higher representation of accepted papers: 39.8% of the responses pertained to papers that were eventually accepted. Out of the 4,435 responses (author-paper pairs) in this phase, 3,259 were from authors who self-identified as male, 310 from authors who self-identified as female, and 866 from those who did not provide a gender or those who did not self-identify as male or female. In terms of participation in the review process, 324 responses were from authors who were invited to serve as meta-reviewers, 1,544 from authors who were invited to serve as reviewers, and 2,567 from neither.

## 5 Main analysis and results

We now present the main results.

### 5.1 Calibration in prediction of acceptance

We begin by looking at the responses to the acceptance probability question, and comparing it with actual acceptance decisions. In Fig 1a, we plot the relation between responses given by authors to the question and the actual acceptance rates for these papers. Here, the blue dots represent responses with at least 50 samples, which together comprise 94% of all responses.

We find that there is a nearly three-fold overestimation overall: The median of the acceptance probabilities estimated by the respondents is 70% and the mean is 67.7%. In comparison, we had primed respondents by mentioning that the acceptance rate in the past four years was about 21%. (The acceptance rate at NeurIPS 2021 ended up being 25.8%.) The fact that participants over-predict aligns with studies in other settings [41, 42] that also find overconfidence effects. Also observe in Fig 1a that interestingly, the authors' predictions track perfect calibration quite well for responses up to 35%, whereas responses greater than (to the right of) 35% are uncorrelated with the actual acceptance rate.

In Fig 1b, we sort the responses in descending order (on the x axis) and plot the values of these responses (y axis). We make separate plots for papers that were eventually accepted and those that were eventually rejected. We see that these two plots track each other quite closely,

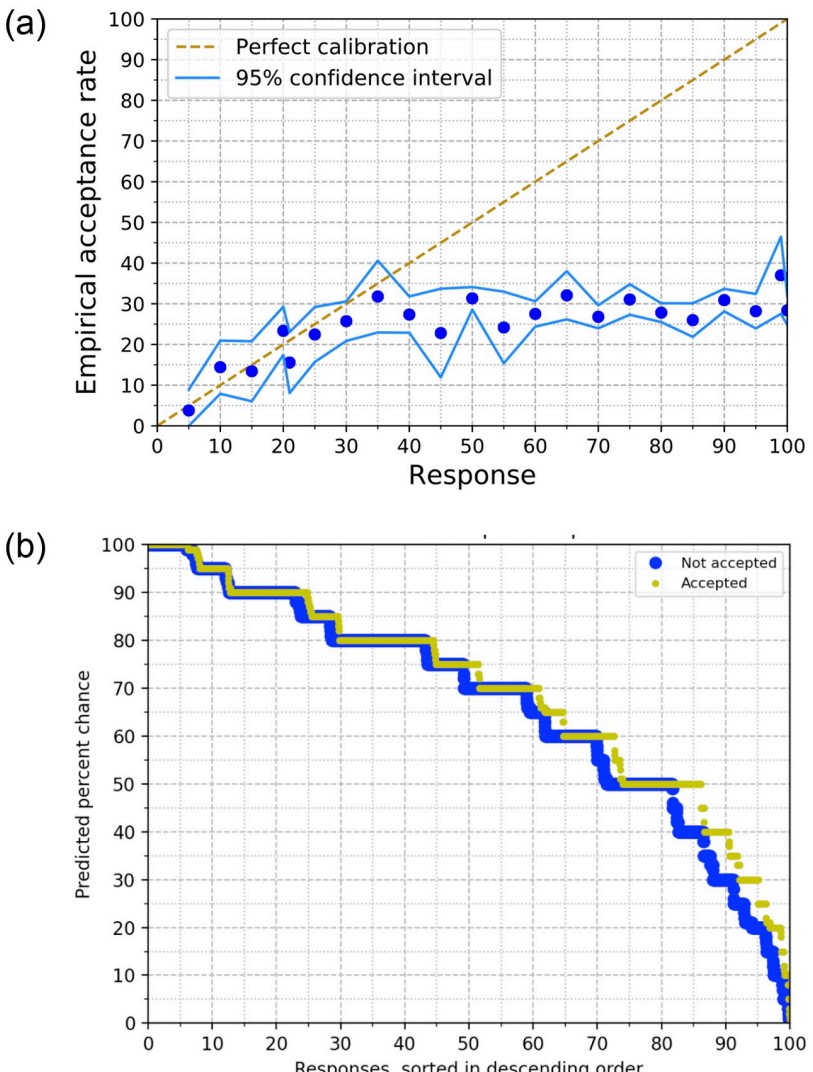

**Fig 1. Author's predictions on the probability of acceptance of their papers.** (**a**) Plot of authors' predictions of chances of acceptance of the paper versus the actual acceptance rates for each response. The diagonal line represents perfect calibration and the (blue) dots represent authors' responses. (**b**) Plots of authors' predictions, for papers that were eventually accepted (thin yellow line) and rejected (thick blue line). The x-axis represents the fraction of responses with predicted percent chance greater or equal to the corresponding value on the y-axis. In other words, the x-axis is the fraction of responses with prediction greater than or equal to the corresponding y value.

with papers that were eventually accepted having slightly higher predictions. We also observe indications of over estimation here—more than 5% of responses predict a 100% chance of their paper getting accepted, about 50% responses predict chances of 75% or higher, whereas fewer than 15% of responses provide a prediction smaller than 40%.

## 5.2 Role of demographics

Next we look at the role of demographics in calibration. For this we now define the calibration error in prediction of acceptance by any author. First, based on Section 5.1 and Fig 1a, we note that responses were on average overly confident, that is the predicted probability of acceptance

was higher than the observed rate of acceptance. Further, we also observe that within each demographic-based subgroup, authors on average predicted a higher acceptance probability of their submission as compared to the acceptance rate within that subgroup. We thus know the direction of miscalibration of each subgroup.

We measure the calibration error of any subgroup in terms of the mean Brier score (i.e., squared loss). The Brier score [43] is a strictly proper scoring rule that measures the accuracy of probabilistic predictions: Given a prediction (value in the interval [0, 1] representing the probability of acceptance) and the outcome (accept = 1, reject = 0), the Brier score equals the square of the difference between the prediction and the outcome. To get a sense of the value of the Brier score, if 25% of the papers are accepted and all respondents provide a prediction of 0.25, then the Brier score equals 0.1875; if all respondents provide a prediction of 0.8 then the Brier score equals 0.49. In our analysis, we had decided in advance to execute statistical tests comparing calibration of male and female authors and of reviewers and meta-reviewers; we had decided to not compare the remaining subgroups due to possibility of high heterogeneity among them. We provide the main details about our analysis in this subsection, and provide additional details in Appendix B.

**5.2.1 Gender.** We compute the average calibration error for gender subgroups using the gender data retrieved from participants' Open Review profiles. Our approach accounts for potential confounding by seniority and geographic region in the analysis for calibration error across gender groups. The calculation is described in detail in Appendix B. See Fig 2a for the average calibration error for the "male", "female" and "not reported" subgroups, where "not reported" comprises of authors who did not provide their gender information in their Open Review profile. We do not report statistics for other gender subgroups, which are very small, to preserve respondent privacy.

In many fields of science there is research showing that there exists a confidence gap between female and male participants [44, 45], where men are generally found to overestimate and women underestimate. In NeurIPS 2021, we test for significance of difference in calibration error by male authors and female authors. To test this hypothesis, we consider the test

(a) (b)

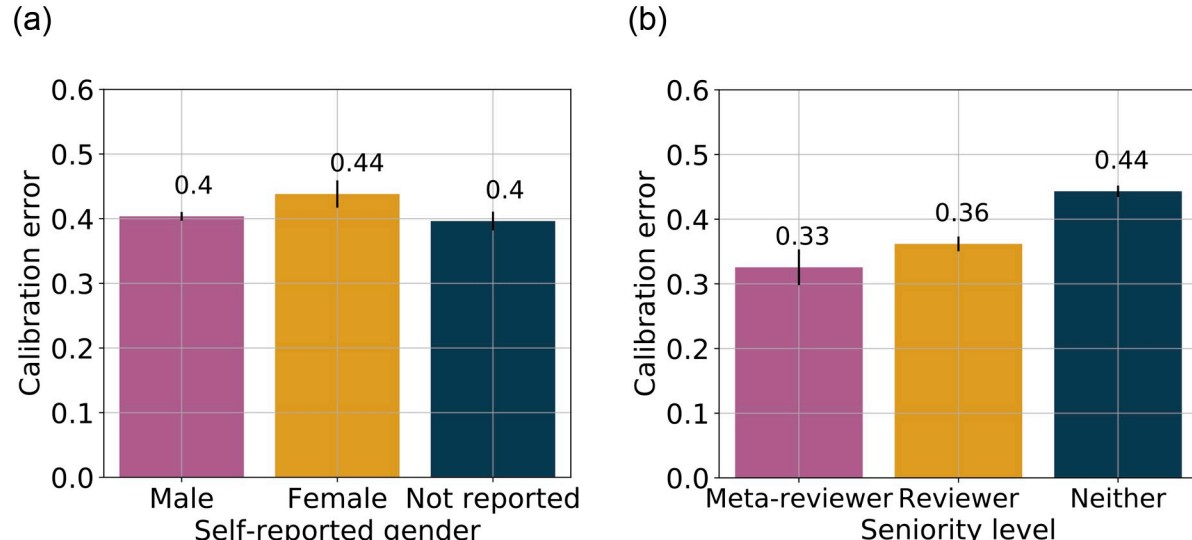

**Fig 2. Comparing authors' calibration error (Brier score) in prediction of acceptance across different subgroups based on gender and seniority level.** The error bars indicate 95% confidence intervals, obtained via bootstrapping. (**a**) Gender-based grouping. (**b**) Seniority-based grouping.

statistic of the difference in calibration errors between female authors and male authors and conduct a two-sided test. We find that there is a statistically significant difference ($p = 0.0012$) at level 0.05. However, note that the effect size—the difference in the calibration errors between female authors (0.44) and male authors (0.40)—is small (0.04).

**5.2.2 Seniority.** We now investigate the role of seniority in authors' calibration of probability of acceptance. As mentioned in Section 4, we consider three subgroups defined by the authors' reviewing role as a proxy for seniority, namely, authors invited to serve as meta-reviewers, authors invited to serve as reviewers, and the remaining authors. Fig 2b shows the average calibration error for these three subgroups, weighted to account for confounding by other demographics (see Appendix B for details). Further, we test for significance of difference in the average calibration error between the sets of meta-reviewers and reviewers. As in Section 5.2.1, we consider the difference in the mean calibration error as the test statistic. The difference in calibration error between meta-reviewers (0.33) and reviewers (0.36) is 0.03, and the difference is not statistically significant ($p = 0.055$) at level 0.05. As mentioned earlier, we had apriori decided to not run any statistical tests on the "neither" group.

## 5.3 Prediction of acceptance vs. perceived scientific contribution

We investigate the consistency between the predictions by authors about the acceptance of their papers and the scientific contribution (paper quality) of those papers as perceived by the authors. There were a total of 6,024 pairs of papers by the same author where the author provided their responses for both questions for both papers. We break down the responses in Fig 3.

Of particular interest are the first two bars in Fig 3 that comprise responses where the same author provided a strict ranking of two papers they authored in terms of their perceived quality, and also gave distinct probabilities of acceptance for the two papers. Among these responses, we find that there is a significant amount of agreement—the two rankings agree in 92.6% $\left(\frac{66.9}{66.9+5.3}\right)$ of responses. However, there is a noticeable 7.4% of responses where the authors thought that the peer review process is more likely to reject the better of their two papers.

## 5.4 Agreements between co-authors, and between authors and peer-review decisions

We first look at author-provided rankings of their perception of the scientific contribution (paper quality) of multiple papers they authored. We compare these rankings with the

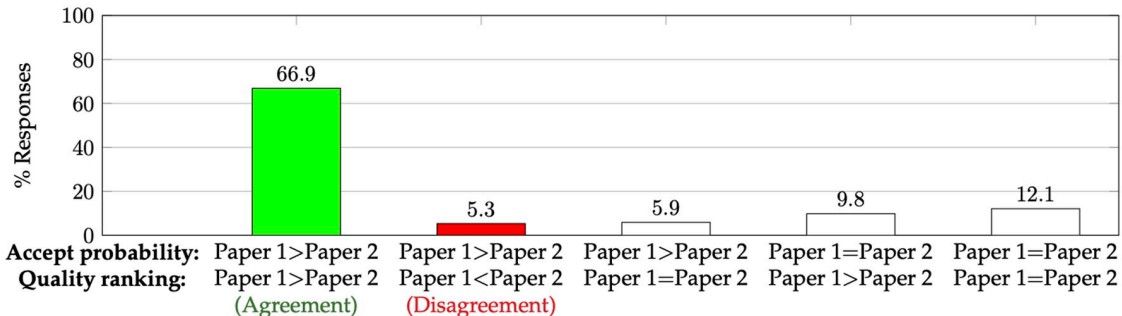

**Fig 3. Comparing authors' (relative) predicted acceptance probability and perceived paper quality for any pair of papers authored by them.** This plot is based on 6,024 such responses. In particular, the first two bars enumerate the amount of agreement and disagreement respectively, among responses of any author that had a strict ranking between the two papers for both questions.

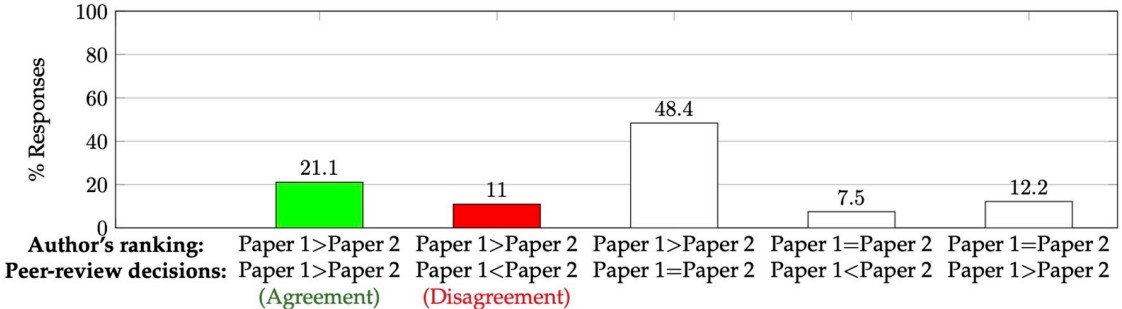

**Fig 4. Comparing authors' ranking of their perceived scientific contribution (paper quality) and the decisions from the peer-review process.** This plot is based on 10,171 such responses. In particular, the first two bars enumerate the agreement and disagreement when the author-provided ranking is strict and where one of the papers is accepted and the other is rejected.

outcomes (accept or reject) of the peer-review process. We show the results in Fig 4. In particular, observe that among the situations where the decisions for the two papers were different and the author-provided ranking was strict (first two bars of Fig 4), authors' rankings disagreed with the decision 34% $\left(\frac{11}{21.1+11}\right)$ of the time. (An analysis comparing the ranking of papers by authors' perceived acceptance probabilities and the final decisions yields results very similar to that in Fig 4.)

We now compute agreements between co-authors in terms of their perceived scientific contribution (paper quality) of a pair of jointly-authored papers. We show the results in Fig 5. Observe that interestingly, among the pairs where both authors gave a strict ranking, they disagreed 32% $\left(\frac{19.7}{41.8+19.7}\right)$ of the time—approximately the same level of disagreement as we saw between the authors and reviewers.

This high amount of disagreement between co-authors about the scientific contribution of their jointly authored papers has some implications for research on peer review. Many models of peer review [21, 38, 46–50] assume existence of some "true quality" of each paper. This result raises questions about such an assumption—if there were such a true quality, then it is perhaps the authors who would know them well at least in a relative sense, but as we saw above, authors do not seem to agree. In a recent work, [51] proposes a novel idea of asking each author to submit a ranking of their submitted papers. Under the assumption that this author-reported ranking is a gold standard, [51] then proposes to modify the review scores to

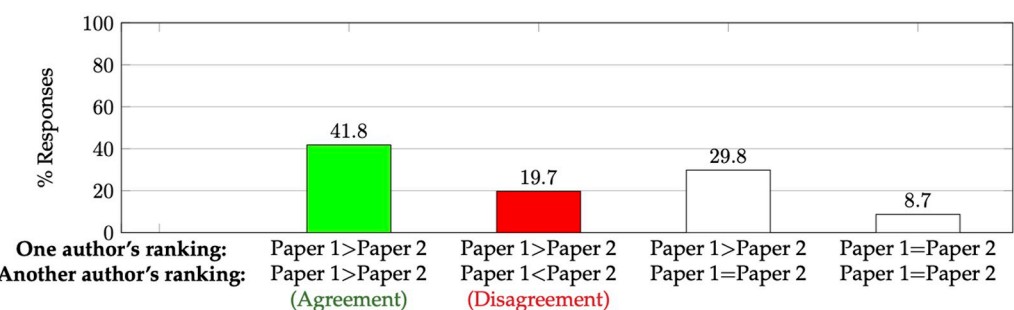

**Fig 5. Comparing co-authors' rankings of their perceived scientific contribution (paper quality) of a pair of papers that both have authored.** This plot is based on 1,357 such responses. In particular, the first two bars enumerate the agreement and disagreement of co-authors when they both provide strict rankings of their papers.

align with this reported ranking. However, our observation that co-authors have a high disagreement about this ranking violates the gold standard assumption that underlies this proposal.

### 5.5 Change of perception

We now analyze the responses to the question posed to authors in the second phase of the experiment on whether the review process changed their perception of their own paper(s). We plot the results in Fig 6. Given significant non-response bias in this phase with respect to acceptance decisions (Section 4), we also separately plot the responses pertaining to accepted and rejected papers.

We observe that among both accepted and rejected papers, about 50% of the responses indicated a change in their perceived opinion about their own papers. Furthermore, even among rejected papers, over 30% of responses mentioned that the reviews made their perception more positive. While past studies [11, 17–19] document whether the review process helps improve the paper, the results in Fig 6 shows that it also results in a change of perception of authors about their papers about half the time.

## 6 Limitations and discussion

This work highlights the perception gap in the peer-review process, between authors, co-authors, and reviewers. Through surveys conducted in NeurIPS 2021, we found that authors considerably overestimated the chances of their papers getting accepted. Our analysis also shed light on the difference in estimation error across demographic-based subgroups. Next, we considered differences in perception across reviewers and authors to find that authors disagreed with peer-review decisions roughly 30% of the times, and co-authors disagreed with each other roughly 30% of the times. Finally, more than 30% of all survey respondents noted that their perception of their submission improved after the review process.

We discuss some key limitations. The 26.7% response rate in phase 1, and particularly the 12.3% response rate in phase 2, introduces concerns about non-response bias, in which non-respondents might have given different answers than respondents. We provide statistics pertaining to non-response bias in Section 4, and attempt to mitigate confounding with respect to

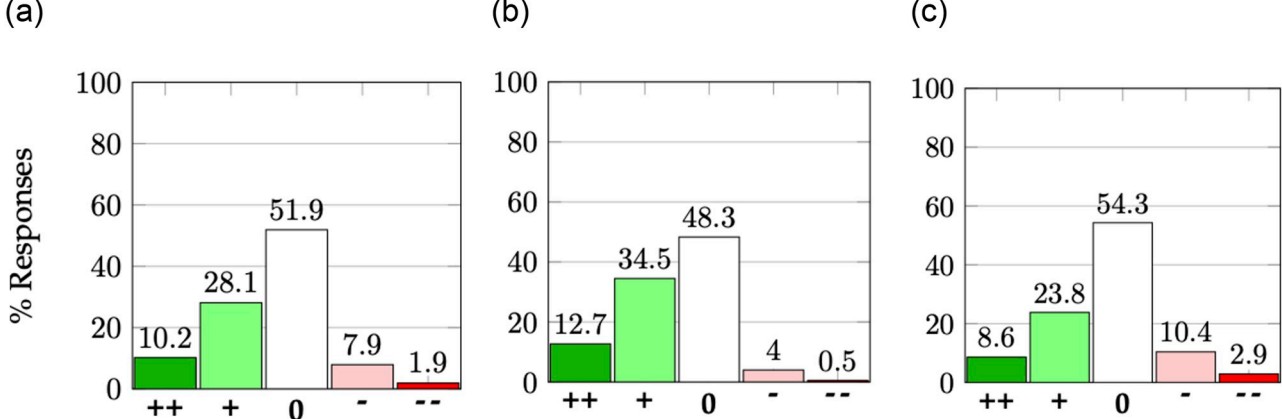

**Fig 6. Change in authors' perceptions of their own papers after seeing the reviews.** The five bars in each plot represent the five options: much more positive ("++"), slightly more positive ("+"), did not change ("0"), slightly more negative ("-"), much more negative ("- -"). The three subfigures depict responses pertaining to all, accepted, and rejected papers, and are based on 4435, 1767, and 2668 such responses respectively. (**a**) All papers. (**b**) Accepted papers. (**c**) Rejected papers.

the observables of demographics and paper outcomes (specifically, Section 5.2 and Section 5.5). However, importantly, only the observables cannot capture all the ways in which data may be missing not at random and this caveat ought be kept in mind in interpreting our results. A second limitation of this study is that respondents may not have been answering fully honestly. For example, if respondents believed that there was even a small chance their answers might leak to reviewers or co-authors, this would incentivize them to exaggerate the probability their paper would be accepted (an effect which would indeed be consistent with the pattern we observed). We took pains to mitigate this effect by assuring the authors of the privacy and security of their responses, and further, by asking them to not discuss their responses with others (see Appendix A).

These limitations notwithstanding, this study has several implications for improving the peer review process. First, the fact that authors vastly overestimated the probability their papers would be accepted suggests there is need for recalibration of expectations. Importantly, peer-review outcomes have been known to impact the graduate school experience of PhD students, with repeated rejections an oft-cited cause for worsening mental health [52]. We hope that the findings in this work will help in cultivating awareness among the student research community about their perceptions as a whole, and facilitate discussions between students and advisors about setting appropriate expectations from the peer-review process, thereby fostering a healthier relationship with their work.

The disagreements we document around paper quality—between co-authors as well as between authors and reviewers—suggest that, as previous work has also found, assessing paper quality is an extremely noisy process. A complementary study on the consistency of decisions made by independent committees of reviewers that was also run at NeurIPS 2021 also showed high levels of disagreement between reviewers [16]. Specifically, 10% of submitted papers were assigned to two independent committees (reviewers, area chairs, and senior area chairs) for review, and of these papers, the committees arrived at different acceptance decisions for 23%. While it may be tempting to attribute this disagreement solely to flaws in the peer-review process, if even co-authors—who know their own work as well as anyone—have significant disagreements on the ranking of their papers, perhaps it is fundamentally hard or impossible to objectively rank papers.

The outcomes of paper submissions should thus be taken with a grain of salt, mindful of the inherent randomness and arbitrariness of the process and the arguable lack of a fully objective notion of paper quality. Realizing that the rejections which generally follow paper submissions do not necessarily result from lack of merit, but rather just bad luck and subjectivity, would both be accurate and healthy for the academic community. More broadly, as a community, we may take these findings into account when deciding on our policies and perceptions pertaining to the peer-review process and its outcomes. We hope the results of our experiment encourage discussion and introspection in the community.

## 7 Appendices

### A More details about the experiment

In this section, we provide details about the experiment, augmenting the details provided in Section 3. First, we focus on the release timeline of the surveys. Then we provide details about the content of the surveys, including the instructions provided.

**Timeline**. Phase 1 of the experiment was conducted soon after the paper submission deadline in order to obtain authors' perceptions of their submitted papers while the papers were still fresh on their minds. The paper submission deadline was on May 28, 2021. The Phase 1 survey was released shortly after, on June 1. Authors were invited to participate in the survey

through June 11, after which the survey was closed. To increase participation in the survey, the program chairs sent a reminder email about the experiment on June 9.

Phase 2 of the experiment aimed at understanding the change in authors' perception of their papers after receiving the initial reviews. The authors received the initial set of reviews on August 3, 2021 and were able to provide their rebuttal (response to the initial reviews) any time until August 10. We invited authors to participate in the Phase 2 survey on August 12. The peer review process was concluded on September 28, 2021, with the announcement of final decisions.

**Instructions**. In both the Phase 1 and Phase 2 surveys, authors were provided information regarding the privacy and confidentiality of their survey responses. They were informed that during the review process, only the authors themselves could view their responses, in addition to the administrators of OpenReview.net (the conference management platform used by NeurIPS 2021). It was emphasised that authors' responses could not affect the outcome of the review process and that the responses would not be visible to their co-authors, reviewers, area chairs, or senior area chairs at any point of time. Regarding the analyses and following dissemination of the findings from the experiment, the survey mentioned that, "After the review process, the survey responses will be made available to the NeurIPS 2021 program chairs and Workflow chairs for statistical analyses. Any information shared publicly will be anonymized and only reported in an aggregated manner that protects your identities." For the purposes of analysis, responses and profiles were accessed algorithmically via the OpenReview api. Further, authors were also told, "To allow authors to freely provide their opinions and keep samples as independent as possible, please do not discuss your answers to these survey questions with other NeurIPS 2021 authors (including your co-authors), or ask others about their responses."

In Phase 1 of the experiment, we asked authors with multiple submissions to rank their submissions. The instructions for providing ranking were as follows: "Rank your papers in terms of your own perception of their scientific contributions to the NeurIPS community, if published in their current form. Rank 1 indicates the paper with the greatest scientific contribution; ties are allowed, but please use them sparingly. In the table entry for each submission below, there is a pull-down menu called "Paper Ranking." Please click on it and specify the rank for that submission."

Finally, among the 6237 authors with multiple submissions, 32 authors (0.5%) provided a ranking for only one of their submissions. We exclude these responses from the analysis of the ranking.

## B More details about demographic analysis

In this section, we provide details about the analyses we conduct to test for significant difference in calibration error across demographic groups in Section 5.2. To describe the analysis, we first define some notation. Let $n$ denote the total number of responses obtained in Phase 1 of our experiment. We will use $i$ as an index over responses, where each response pertains to a single author-paper pair. For response $i$, let $p_i \in [0, 1]$ be the acceptance probability indicated by the author. The observed outcome of the associated paper is a binary indicator, denoted by $y_i \in \{0, 1\}$, where $y_i = 1$ if the paper is accepted and $y_i = 0$ if it is rejected. The self-reported gender of the associated author is denoted by $g_i \in G \coloneqq \{\text{Female, Male, Other, Unspecified}\}$. Note that there are responses where the associated authors did not provide a gender in their Open Review profile. All authors' seniority is classified into three types based on their reviewing participation, denoted by $s_i \in S \coloneqq \{\text{Meta-reviewer, Reviewer, Neither}\}$.

Finally, we include the geographical region associated with the author, denoted by $r_i$. To assign a geographical region to each author, we use the institutional domain of the author's

primary affiliation. We classify the geographical regions using the geographical region division provided by the United Nations Statistics Division (UNSD). Within their division of regions, we further break each region with more than 100 responses in our survey into sub-regions listed by UNSD. This yields the following set of regions denoted by $R := \{$Africa, North America, Latin America and the Caribbean, Central Asia, Eastern Asia, South-eastern Asia, Southern Asia, Western Asia, Eastern Europe, Northern Europe, Southern Europe, Western Europe, Oceania$\}$.

To measure accuracy, we use the Brier score (i.e., squared loss). For response $i$, the Brier score is given by $(y_i - p_i)^2$. With this notation, we define the average calibration error for a gender-based subgroup. To account for confounding by authors' seniority and geographical region, we bin all responses based on their corresponding seniority and geographical region, and compute their prevalence rate in the population. This gives the weight to be assigned to each response to compute the average calibration error for gender-based subgroups as

$$M_g = \sum_{r \in R} \sum_{s \in S} \left( \frac{\sum_{i \in [n]} \mathbb{I}(g_i = g, r_i = r, s_i = s)(y_i - p_i)^2}{\sum_{i \in [n]} \mathbb{I}(g_i = g, r_i = r, s_i = s)} \times \frac{\sum_{i \in [n]} \mathbb{I}(r_i = r, s_i = s)}{n} \right), \qquad (1)$$

where $\mathbb{I}(\cdot)$ is the indicator function. Using this definition of calibration error of a gender subgroup, we derive 95% confidence intervals using bootstrapping [53]. We now move on to our hypothesis comparing miscalibration between male authors and female authors. Formally, in terms of (1), the hypothesis is stated as:

$$H_0 : M_{\text{male}} = M_{\text{female}},$$

$$H_1 : M_{\text{male}} \neq M_{\text{female}}.$$

To test this hypothesis, we conduct a permutation test to obtain its significance ($p$-value). In the permutation test, we permute our data within each demographic subgroup of seniority and geographical region. From the permutation test, we obtain a $p$-value of 0.0006. To account for multiple testing we use the Benjamini-Hochberg procedure [54], which gives a final $p$-value of 0.0012.

Similarly, we compute the average calibration error for seniority-based subgroups, while accounting for confounding by gender and geographical region. In this analysis, we filter out the responses by authors who did not report their gender. Since the set of authors who did not report their gender may be a heterogeneous set, including this set in the analysis for seniority will violate the exchangeability assumption of the permutation test. Thus, the total number of responses considered in the seniority analysis, denoted by $n_{g \in G}$, is given by $\sum_{i \in [n]} \mathbb{I}(g_i \in G)$. With this, the average calibration error corresponding to each seniority level, for $s \in S$, is given by

$$M_s = \sum_{r \in R} \sum_{g \in G} \left( \frac{\sum_{i \in [n]} \mathbb{I}(s_i = s, r_i = r, g_i = g)(y_i - p_i)^2}{\sum_{i \in [n]} \mathbb{I}(s_i = s, r_i = r, g_i = g)} \times \frac{\sum_{i \in [n]} \mathbb{I}(r_i = r, g_i = g)}{n_{g \in G}} \right). \qquad (2)$$

We use bootstrapping to compute 95% confidence intervals. Further, we conduct a permutation test to compare the miscalibration by meta-reviewers (ACs and SACs) and other reviewers. This hypothesis is stated as

$$H_0 : M_{\text{meta-reviewer}} = M_{\text{reviewer}},$$

$$H_1 : M_{\text{meta-reviewer}} \neq M_{\text{reviewer}},$$

where $M_{\text{meta-reviewer}}$ and $M_{\text{reviewer}}$ are as defined in (2). The permutation test yields a $p$-value of 0.055. Accounting for multiple testing using the Benjamini-Hochberg procedure does not alter the $p$-value.

## Acknowledgments

We would like to thank all the participants for the time they took to provide survey responses. We are grateful to the OpenReview team, especially Melisa Bok, for their support in running the survey on the OpenReview.net platform.

## Author Contributions

**Conceptualization:** Charvi Rastogi, Ivan Stelmakh, Alina Beygelzimer, Yann N. Dauphin, Percy Liang, Jennifer Wortman Vaughan, Hal Daumé III, Emma Pierson, Nihar B. Shah.

**Data curation:** Charvi Rastogi, Ivan Stelmakh, Alina Beygelzimer, Yann N. Dauphin, Percy Liang, Jennifer Wortman Vaughan, Zhenyu Xue, Hal Daumé III, Emma Pierson, Nihar B. Shah.

**Formal analysis:** Charvi Rastogi, Alina Beygelzimer, Zhenyu Xue, Hal Daumé III, Emma Pierson, Nihar B. Shah.

**Investigation:** Charvi Rastogi, Alina Beygelzimer, Yann N. Dauphin, Percy Liang, Jennifer Wortman Vaughan, Zhenyu Xue, Hal Daumé III, Emma Pierson, Nihar B. Shah.

**Methodology:** Charvi Rastogi, Ivan Stelmakh, Alina Beygelzimer, Yann N. Dauphin, Percy Liang, Jennifer Wortman Vaughan, Zhenyu Xue, Hal Daumé III, Emma Pierson, Nihar B. Shah.

**Project administration:** Jennifer Wortman Vaughan.

**Resources:** Alina Beygelzimer.

**Supervision:** Alina Beygelzimer, Yann N. Dauphin, Percy Liang, Jennifer Wortman Vaughan, Hal Daumé III, Emma Pierson, Nihar B. Shah.

**Validation:** Charvi Rastogi, Alina Beygelzimer, Jennifer Wortman Vaughan, Zhenyu Xue, Hal Daumé III, Emma Pierson, Nihar B. Shah.

**Visualization:** Charvi Rastogi, Alina Beygelzimer, Jennifer Wortman Vaughan, Zhenyu Xue, Hal Daumé III, Emma Pierson, Nihar B. Shah.

**Writing – original draft:** Charvi Rastogi, Ivan Stelmakh, Alina Beygelzimer, Jennifer Wortman Vaughan, Emma Pierson, Nihar B. Shah.

**Writing – review & editing:** Charvi Rastogi, Alina Beygelzimer, Jennifer Wortman Vaughan, Hal Daumé III, Emma Pierson, Nihar B. Shah.

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
