## [Decision Letter · Decision Letter 0]

8 Feb 2024

PONE-D-23-39630How do Authors’ Perceptions of their Papers Compare with Co-authors’ Perceptions and Peer-review Decisions?PLOS ONE

Dear Dr. Shah,

Thank you for submitting your manuscript to PLOS ONE. After careful consideration, we feel that it has merit but does not fully meet PLOS ONE’s publication criteria as it currently stands. Therefore, we invite you to submit a revised version of the manuscript that addresses the points raised during the review process.

We look forward to receiving your revised manuscript.

Kind regards,

Tobias Otterbring

Academic Editor

PLOS ONE

Journal Requirements:

**Additional Editor Comments:**

Dear authors,

Your paper has now been reviewed by two scholars with considerable knowledge and expertise in this topic domain. Both the reviewers like many aspects of your work and only suggest minor areas of improvement (please see their detailed comments below or in attached files). Based on my own reading of your manuscript, I concur with the reviewers' positive evaluation of your research. Accordingly, I am happy to invite you for a minor revision, with a clear path toward publication as long as you carefully address all thoughtful comments from the reviewers.

Kind regards,

Tobias Otterbring

Handling Editor, PLOS One

Reviewers' comments:

Reviewer's Responses to Questions

**Comments to the Author**

1. Is the manuscript technically sound, and do the data support the conclusions?

Reviewer #1: Yes

Reviewer #2: Yes

2. Has the statistical analysis been performed appropriately and rigorously? 

Reviewer #1: Yes

Reviewer #2: Yes

3. Have the authors made all data underlying the findings in their manuscript fully available?

Reviewer #1: No

Reviewer #2: Yes

4. Is the manuscript presented in an intelligible fashion and written in standard English?

Reviewer #1: Yes

Reviewer #2: Yes

5. Review Comments to the Author

Reviewer #1: This paper confirms empirically what has been anecdotally known, that authors have a much higher expectation for acceptance of their work, even for venues where acceptance rates are very low. I applaud the authors of this paper for confirming this at a very large scale. As the authors note, there could be selection bias, and it is difficult to tell whether authors who simply did not opt into the survey had higher or lower expectations of acceptance.

It is interesting to note that authors who were invited to be metareviewers and reviewers were better calibrated. This is also not a surprising result, as these are senior authors who are both experienced and also less invested in each submission (compared to more junior authors who were probably main authors rather than corresponding).

I am a bit concerned about the gender difference reported, as there could be intersectionality here. Are there more senior male authors than senior female authors? In that case, would it make sense to control for seniority?

Another concern is about the data release. I understand that compromising reviewer confidentiality would be problematic, but it would be nice to share this dataset somehow with the community.

Overall, this will add to the body of work studying the peer review process and hopefully lead to improvements in the process.

Reviewer #2: This is an interesting and well designed study. I have only a couple of minor comments.

1. Some additional proofreading is needed. Most of the verbs were stated in present tense when they should have been past tense.

2. The overall findings should be summarized in the conclusion, then the limitations should be discussed.

3. It might be useful to speculate about how the problem of overestimation of the likelihood of acceptance impacts the trust in peer review. Could it lead to dissatisfaction?

6. PLOS authors have the option to publish the peer review history of their article (what does this mean?). If published, this will include your full peer review and any attached files.

Reviewer #1: No

Reviewer #2: **Yes: **David B Resnik

---

## [Author Response · Author response to Decision Letter 0]

1 Mar 2024

Please see attached file on response to reviewers. Thank you.

---

## [Editor Report · Decision Letter 1]

5 Mar 2024

How do Authors’ Perceptions of their Papers Compare with Co-authors’ Perceptions and Peer-review Decisions?

PONE-D-23-39630R1

Dear Dr. Shah,

We’re pleased to inform you that your manuscript has been judged scientifically suitable for publication and will be formally accepted for publication once it meets all outstanding technical requirements.

Kind regards,

Tobias Otterbring

Academic Editor

PLOS ONE

Additional Editor Comments (optional):

Dear authors,

Based on my reading of your revision and the rather minor aspects you were requested to improve according to the former feedback provided by the reviewers, my assessment is that your paper now meets the standards for publication in this journal. For that reason my recommendation as the handling editor is to accept your paper for publication in PLOS One.

Kind regards,

Tobias Otterbring

---

## [Editor Report · Acceptance letter]

20 Mar 2024

PONE-D-23-39630R1 

PLOS ONE

Dear Dr. Shah, 

I'm pleased to inform you that your manuscript has been deemed suitable for publication in PLOS ONE. Congratulations! Your manuscript is now being handed over to our production team.

Kind regards, 

on behalf of

Professor Tobias Otterbring 

Academic Editor

PLOS ONE